# Enhancing Salty Taste Perception in Stroke Patients via Anodal Electrical Stimulation to the Chin

**DOI:** 10.3390/foods13244087

**Published:** 2024-12-17

**Authors:** Masahito Katsuki, Taiki Fukushima, Naomichi Wada, Tetsuya Goto, Ayana Imai, Yasuko Hanaoka, Takuji Yasude, Kazuma Kaneko, Tetsuyoshi Horiuchi

**Affiliations:** 1Physical Education and Health Center, Nagaoka University of Technology, Nagaoka 940-2137, Japan; 2Department of Neurosurgery, Japanese Red Cross Society Suwa Hospital, Suwa 392-0027, Japan; 3UBeing, Inc., Nagoya 453-6111, Japan; 4Department of Neurology, Japanese Red Cross Society Suwa Hospital, Suwa 392-0027, Japan; 5Department of Neurosurgery, Shinshu University School of Medicine, Matsumoto 390-8621, Japan

**Keywords:** anodal electrical taste stimulation, hypertension, rehabilitation, salt reduction, stroke, taste manipulation

## Abstract

A lower salt intake is an effective management strategy for hypertension and ultimately stroke. However, this strategy compromises the taste of food. To overcome this, a taste manipulation strategey using electronic taste simulation (ETS) has been established, but this has only been studied in healthy individuals. Therefore, this study aimed to demonstrate and quantitatively evaluate the taste enhancement effect of ETS in patients admitted to a hospital due to stroke. Twenty patients (mean = 67.8 ± 13.6 years) underwent two psychophysical experiments to assess the effects of ETS on salt taste perception using salt-impregnated filter paper. The patients’ stroke types included twelve ischemic and eight hemorrhagic strokes. The median salt taste thresholds without ETS and with ETS were 0.7% and 0.6%, respectively (*p* = 0.083). The perceived concentration for the 0.8% concentration increased from 0.8% to 1.0% with the ETS (*p* = 0.041), and for the 1.0% concentration, from 1.0% to 1.2% (*p* < 0.001). The findings suggest that ETS significantly enhances salty taste perception in patients who have experienced a stroke without altering salt concentration, potentially aiding in reducing daily salt intake. Further research is necessary to explore its broad applicability in dietary management and blood pressure control.

## 1. Introduction

Hemorrhagic and ischemic strokes are the two main causes of stroke and are associated with high morbidity and mortality. Moreover, high blood pressure (BP), also referred to as hypertension, is the most prevalent risk factor for stroke; therefore, lowering BP and maintaining a target BP of <130/80 mmHg is the most effective way to prevent both initial and recurrent strokes [1,2].

High salt intake leads to hypertension by increasing blood volume through fluid retention, narrowing blood vessels, and impairing kidney function. Excess sodium activates hormonal systems like the renin–angiotensin–aldosterone system, causing the further constriction of blood vessels. These combined effects result in elevated blood pressure, contributing to the development of hypertension [3].

A lower salt intake is an important nonpharmacological approach for managing hypertension [4]. According to the Japanese Society of Hypertension Guidelines for the Management of Hypertension, an individual should have a daily salt intake of less than 6 g [3]. A reduction of 1 g of dietary salt daily can lower an individual’s BP by 1 mmHg [4]. However, this reduction often results in an unsatisfactory taste. Hence, identifying methods to reduce salt intake without sacrificing flavor is essential.

Electrical taste stimulation (ETS) has been increasingly recognized for its potential to improve sensory perception, particularly in the context of reducing salt intake without compromising flavor [5,6,7]. Previous studies have explored various forms of electrical stimulation, including electrogustometry (EGM), to assess and enhance taste perception [8,9]. Electrogustometry, which involves the direct stimulation of taste receptors on the tongue, has been shown to alter the perceived intensity of tastes, particularly in individuals with sensory deficits such as those following stroke [2]. ETS, in comparison, often uses anodal stimulation, which has been shown to improve sensory thresholds in various regions of the oral cavity, including the faucial pillars, which are crucial for triggering the swallowing reflex [10,11]. The key mechanism behind ETS is believed to involve the activation of peripheral sensory nerves, which then modulate central nervous system activity, enhancing sensory processing and potentially improving taste perception without the need for altering the chemical composition of food. This ability to modulate taste perception could be particularly beneficial for individuals seeking to reduce their salt intake, as it may allow them to enjoy food with lower sodium content while still experiencing satisfactory flavor. While the use of sour and cold sensory stimuli in stroke rehabilitation has demonstrated positive effects on swallowing function and pharyngeal sensory input [11], the potential of ETS in reducing salt intake without compromising flavor remains underexplored. This study aims to investigate the effects of anodal ETS on salt taste perception in stroke patients, exploring its potential as a non-invasive intervention to support salt reduction strategies in this population.

Anodal ETS addresses the challenge of balancing health with taste. ETS causes changes in taste by directly stimulating sensory nerves and changing the localization of taste substances [12]. This technique allows individuals to enjoy healthy foods while experiencing satisfactory flavors without increasing the concentration of taste components. Anodal ETS has some advantages compared to other salt reduction methods: Reducing salt by replacing sodium chloride with potassium chloride cannot be used by patients with poor kidney function due to the potassium load. In addition, the cutlery-type ETS device must be kept in contact with the mouth during chewing in order to be effective. While the effectiveness of ETS in enhancing salt taste perception has been demonstrated in healthy individuals [12] as well as in cases involving subarachnoid hemorrhage (SAH) [5], its impact on patients who experience strokes is yet to be established. Therefore, this study aims to evaluate the effectiveness of an ETS device in improving salt taste perception in patients with stroke.

## 2. Materials and Methods

### 2.1. Study Population and General Management

Previous research determined that 70% of healthy volunteers in their 20s reported a significantly enhanced perception of salt taste when using the same ETS [13]. This study aimed to determine whether more than 70% of stroke patients experience similar results. With a confidence level of 95% and a margin of error of 5%, a sample size of 19 was required. Considering potential dropouts, a study cohort of 20 patients was selected for this study.

From October 2023 to February 2024, 20 consecutive patients experiencing strokes were admitted to the Stroke Care Unit in Japanese Red Cross Society Suwa Hospital and were prospectively enrolled in this study. A stroke neurologist or neurosurgeon conducted a stroke diagnosis on the patients based on clinical history, neurological symptoms, and radiological findings. Treatments, including surgical procedures and intravenous thrombolytic therapy, were performed according to the Japan Stroke Society Guideline 2021 for the Treatment of Stroke [14]. For approximately two weeks, the patients were treated in the intensive care unit or stroke care unit and then transferred to a general ward for subsequent treatment. During hospitalization, the patients also received active nutritional therapy and rehabilitation.

Stroke patients are generally prescribed a special diet for reduced salt. The caloric intake is calculated using the Harris–Benedict equation along with an activity factor, while the salt intake is adjusted to be below 6 g per day.

### 2.2. Inclusion and Exclusion Criteria

Eligible participants for this study were patients aged 20–90 years who had experienced strokes regardless of the presence of hypertension. Those who were pregnant, were suspected of having or currently receiving treatment for epilepsy, or had severe heart disease were excluded from the study. Additionally, those with subjective or objective taste or smell dysfunction, poor health, fever (temperature ≥ 37.5 °C), inflammation, or lesions in or around the mouth were not eligible. Participants who used pacemakers, implantable cardioverter-defibrillators, or deep brain stimulation devices were excluded. Individuals who wore piercings or earrings were included if they could be removed during the trial. Those unable to provide informed consent due to dementia and higher-order dysfunction and those judged to have a poor general condition and would have had difficulty participating in the experiment were also excluded from the study.

### 2.3. Experimentation

After the third week of admission, the patients were transferred to the general ward. Similarly to a previous study, two psychophysical experiments, as described later, were conducted to demonstrate the effect of ETS on taste perception [5]. As shown in Figure 1, the disposable electrodes, 36 mm in diameter, equipped with conductive gel and metal pins for optimal contact (C915F30, Saitama, Chibara, Japan), were placed at the center of the chin (anode) and in the area around the hyoid bone (cathode). These electrodes were directly adhered to the chin using the adhesive (gel). The stimulation wires were standard copper wires (0.2 Sq) connected to the electrodes with crocodile clips. The ETS device was equipped with an ampere meter, and the circuit was designed to allow a constant current to flow using a direct current power supply. The current was gradually increased from 0 mA to 1.0 mA over about 2 s manually and allowed to stabilize before the experiment began. The current was continuously applied throughout the experiment. A constant current was used, rather than a sudden 1 mA spike, to avoid distracting the participants from the taste stimuli.

Previous studies have demonstrated that electrical taste stimulation with the chin as the anode and the neck as the cathode enhances saltiness in healthy individuals [15]. This method, which does not require placing electrodes or cables inside the mouth, is expected to enable virtual taste enhancement without interfering with eating. Additionally, placing the anode over the chin and the cathode over the jaw is presumed to activate taste-related nerves by directing current into the oral cavity. Using this approach, we have previously reported enhanced saltiness in stroke patients [5]. Based on these findings, this study employed electrical taste stimulation with the jaw as the anode and the submandibular region as the cathode.

In Experiment 1, the patients were seated and instructed to assess the salt taste intensity of salt-impregnated test papers (SALSAVE^®^, 07830010, Advantech, Toyama, Japan). SALSAVE^®^ consists of a set of seven filter papers with varying salt concentrations: 0%, 0.6%, 0.8%, 1.0%, 1.2%, 1.4%, and 1.6%. Each filter paper was placed on the patient’s tongue, and their mouths were closed (Figure 2).

First, starting with the paper from the lowest concentration to highest concentration in order, the paper that elicited a salty taste was recorded as the taste threshold. Followed by a thorough mouth rinse and a rest period of at least 10 min, the same procedure was repeated using a direct current of 1.0 mA via the ETS and the salt taste threshold was re-evaluated. Experiment 1 was conducted without disclosing salt concentrations to the patients [16].

Experiment 2 was conducted after the patients thoroughly rinsed their mouths and rested for at least 10 min. The seven concentration levels of SALSAVE^®^ were then disclosed to the patients, and they were required to remember them. The patients were then given 0.8% filter paper without revealing the concentration. They were asked to determine the concentration of the filter papers ranging from 0.6% to 1.6% without ETS. After rinsing their mouth and resting for at least 10 min, the same filter papers were given to the participants with a 1.0 mA direct current applied by the ETS device. The patients were then asked to indicate the concentration they assumed it might be. After rinsing their mouth, the patients were then provided with 1.0% filter paper, and the same procedure was repeated. This experiment was conducted with and without ETS at two different concentrations: 0.8% and 1.0% [17].

The primary evaluation items were the proportions of whether the ETS device decreased the threshold concentration in Experiment 1 and whether the perceived concentration was increased by the ETS device in Experiment 2. The procedure and timeline are described in Figure 3.

### 2.4. Statistical Analysis

The results were presented as median (interquartile range, IQR) for the variables without normal distribution and mean ± standard deviation (SD) for the variables with normal distribution. The Shapiro-Wilk test confirmed a normal distribution. Fisher’s exact test and the chi-squared test were used to compare the proportion of the categorical variables. The Mann–Whitney U test was used to compare continuous and ordinal variables in independent groups. The Wilcoxon signed-rank test was used as a paired test to compare continuous and ordinal variables. The outcomes in experiments 1 and 2 were evaluated using the Wilcoxon signed-rank test. These analyses were conducted using SPSS v28.0.0 (IBM, Armonk, NY, USA), Python v3.9, and matplotlib v3.9.2. Statistical significance was set at *p* < 0.05.

### 2.5. Ethical Aspects

The Japanese Red Cross Society Suwa Hospital Ethics Committee approved this study (approval number: R5-2), and the safety standards used in existing reports were followed [12]. Written informed consent was obtained from all 20 patients before study participation. All methods were performed in accordance with the Strengthening the Reporting of Observational Studies in Epidemiology guidelines and the regulations of the Declaration of Helsinki.

## 3. Results

### 3.1. Patients’ Characteristics

The mean (SD) age of the cohort was 67.8 (13.6) years, and three (14.3%) were female. Fourteen patients (70%) had a history of hypertension and were taking antihypertensive drugs during the experiment. The mean (SD) estimated glomerular filtration rate (eGFR) was 69.9 (18.4) mL/min/1.73 m^2^. The stroke types were ischemic stroke, including eleven cerebral infarctions and one transient ischemic attack, and hemorrhagic stroke, including seven cerebral hemorrhages and one SAH. The median (interquartile range [IQR]) National Institutes of Health Stroke Scale (NIHSS) score at admission was 3 (3) (Table 1). When the experiment was conducted, all patients had clear consciousness, no dementia and/or higher-order dysfunction, and no taste or smell disorders.

### 3.2. Results of Experiment 1

In Experiment 1, the median (IQR) salt-taste thresholds without and with ETS were 0.7% (0.4%) and 0.6% (0.2%), respectively. However, the salt percentage threshold was not significantly reduced (*p* = 0.083 by Wilcoxon signed-rank test). Seven of the twenty patients (35%) showed threshold reductions due to ETS (Table 1 and Figure 4A). The threshold for each case is shown in Figure 4B.

### 3.3. Results of Experiment 2

In Experiment 2, the median (IQR) perceived concentrations for 0.8% salt filter paper without and with ETS were 0.8% (0.1%) and 1.0% (0.3%), respectively. The perceived concentration significantly increased (*p* = 0.041 by Wilcoxon signed-rank test). Eleven of twenty patients (55%) experienced enhanced salt perception (Table 1 and Figure 4C). The perceived concentration in each case for the 0.8% filter paper is shown in Figure 4D.

The median (IQR) perceived concentrations for the 1.0% salt filter paper without and with ETS were 1.0% (0.1%) and 1.2% (0.1%), respectively. The perceived concentration significantly increased (*p* < 0.001 by Wilcoxon signed-rank test). Seventeen of twenty patients (85%) experienced enhanced salt perception (Table 1 and Figure 4E). The perceived concentration in each case for the 1.0% filter paper is shown in Figure 4F.

No adverse events related to ETS were observed. The patients were asked about the discomfort caused by ETS through an experiment using a numerical rating scale ranging from 0 to 10. The median (IQR) was 2 (4), and three (15%) patients reported no discomfort during the experiments.

In Experiments 1 and 2, age, sex, presence of hypertension, eGFR, NIHSS score, and discomfort were compared between patients with and without ETS effects. There were no statistically significant differences between the two groups (*p* = 0.341, 0.207, 0.999, 0.843, 0.547, 0.678, respectively, in Experiment 1. *p* = 0.552, 0.465, 0.785, 0.764, 0.999, 0.557, respectively, in Experiment 2 with 0.8% salt filter paper. *p* = 0.842, 0.779, 0.765, 0.473, 0.237, 0.337, respectively, in Experiment 2 with 1.0% salt filter paper. Fisher’s exact and chi-squared tests for categorical variables and Mann–Whitney U tests for continuous and ordinal variables were used).

## 4. Discussion

Unlike previous studies conducted with healthy volunteers, this report marks the first observation that ETS improves salty taste perception in patients who experience strokes. Here, the mechanisms behind the ETS device and its potential implications for the future treatment of hypertension are discussed.

### 4.1. ETS in Patients Who Experience Strokes: A Comparison with Healthy Volunteers

Nakamura et al. [12] conducted a study on an anodal ETS device involving six healthy university student volunteers to evaluate its salt taste-enhancing effects. They tested various saline solutions, ranging from 1 to 8%, and determined that the lower concentrations had a stronger enhancing effect. The potentiation effect of the ETS was also assessed using direct currents of 1–3 mA, with higher currents resulting in greater enhancement. For example, Sakurai et al. [18] used a continuous square-wave ETS with five healthy volunteers in their 20s to examine the salt taste-enhancing effect, and all participants reported an increase in salt perception with ETS.

Compared with previous reports on healthy volunteers, not all patients within the cohort reported enhanced salt perception. The perception-enhancing effect was more significant on the high-concentration salted filter paper than on the lower-concentration filter paper in Experiment 2. However, age, sex, NIHSS score at admission, and discomfort with ETS did not differ between the groups with and without ETS effects.

Taste perception deteriorates with age. Therefore, slight differences in the salt concentrations may not have been detected. Even if the salt perception was enhanced, the change would not be felt at lower concentrations. Additionally, severe neurological deficits and cerebrovascular diseases in the frontal circulation are risk factors for taste disorders after acute stroke. Post-stroke taste dysfunction occurs in 30% of patients within one year, and approximately half of these improve [19]. The effectiveness of ETS may decrease with age and stroke occurrence; therefore, further investigations are required to determine which patients are most susceptible to it. By contrast, taste disorders can increase salt intake, which is a risk factor for stroke [20]. If ETS can improve taste disorders and lead to lower salt intake, it may reduce the risk of stroke in patients with taste disorders.

### 4.2. Mechanism of ETS

There are various types of ETS. Some use forward or backward [17] currents, as well as alternating or direct currents [21], and the waveform of the current can vary [22]. A simple ETS device was used in this study, in which 1 mA of direct current was passed through the skin to the chin and jaw (Figure 1). Various studies are being conducted to determine what kind of ETS can modify taste; however, the underlying mechanisms are still largely unknown. Previous studies have stated two possible mechanisms by which electrical stimulation from the skin increases or decreases taste.

The first hypothesis suggests that changes in taste occur through the direct stimulation of taste-afferent nerve fibers. Electrical stimulation may directly affect nerve excitation [22]. In other words, the electrical current stimulates the sensory nerves and induces a virtual taste that increases the salty taste of saltwater, giving the illusion of a stronger taste.

The second proposed mechanism is that the electric field created by electrical stimulation creates a concentration gradient by the electrophoresis of taste substances such as sodium ions, creating a pseudo-taste sensation. In the taste conduction pathway, the action of taste substances on channels or receptors on taste cells in the taste buds of the tongue triggers gustatory nerve excitation. Electrical stimulation of the skin creates an electric field that generates a taste concentration gradient. Consequently, certain areas of the tongue have a higher concentration of taste substances than foods and drinks that enter the oral cavity, and we may experience a falsely strong taste [15]. The concept of electric taste has also been proposed as having a unique flavor [12] and virtual taste sense that can detect the taste of salt, even in its absence [23]. Therefore, ETS may act on a process that is different from conventional taste perception, and further research is required.

### 4.3. Comparison of ETS Devices Shaped Like Tableware

The device used in this study applies current directly to the jaw, whereas other ETS devices are designed to resemble tableware. These tableware-type devices can be used naturally without electrodes attached to the body, and their utility has been documented [23].

However, traditional ETS cannot transmit electric currents and enhance taste unless the electrodes and food in contact with them touch the body. Consequently, when using a tableware-like ETS device, taste alteration may only occur when the food and tableware-shaped ETS devices are placed in the mouth. Additionally, because electrical stimulation is applied directly to the tongue and oral mucosa, there is a risk of mucosal damage and the potential for an unpleasant bitter taste.

The advantage of ETS which involves transcutaneous electrical stimulation of the chin and jaw lies in its long-lasting effects and reduced risk of mucosal damage. However, a disadvantage of the ETS device used in this study is the attachment process of the electrodes to the skin, as this process requires time and effort. Further research is required to identify the most effective device.

### 4.4. Limitations and Considerations

The sample size was as small as 20 patients, and the statistical significance of some of the findings is marginal (*p* = 0.041). Therefore, we should confirm ETS’s effects in a larger sample size. The generalizability of these findings should be confirmed not only for patients experiencing strokes within the hospital used in this study, but also for patients with other conditions or those in other hospitals. The use of a single current intensity (1.0 mA) in this study may not be sufficient to fully explore the dose–response relationship. Future studies should consider testing a range of current intensities to determine the optimal stimulation parameters. When considering detailed laboratory or radiological test results, patients who do and do not experience the effects of ETS should be investigated. Also, a potential limitation of this study is the lack of randomization in the order of conditions and the absence of a sham stimulation control group. This could introduce a carryover effect from the initial evaluation to the subsequent round with stimulation. Randomizing the order and including a sham group would have strengthened the design and minimized bias. Additionally, the study evaluated taste perception at only one point, limiting our understanding of the long-term effects of salt intake habits, BP trends over time, and hypertension-related events such as recurrent strokes. Further research is needed to integrate ETS into the dietary management plans of stroke patients and to evaluate ETS’s long-term practicality and cost-effectiveness to ensure it is both beneficial and accessible for stroke patients. The potential side effects of the ETS should also be considered. In addition, objective assessments should be conducted, such as taste tests using electrogustometry and the taste disk method, along with smell tests such as the alinamin test. This is important because taste and smell disorders may be associated with stroke and must be objectively evaluated [19,24]. There are no prior studies that specifically investigate this approach in this population. As such, a detailed comparison with previous research regarding methodological aspects such as population and ETS devices is not feasible at this stage. We plan to address these comparisons and provide a more in-depth analysis in future studies as we expand the understanding of ETS in stroke rehabilitation.

Finally, SALSAVE^®^ is a tool designed to assess saltiness, and this study focused specifically on changes in salt taste perception. However, previous studies [25] in healthy individuals have reported that electrical stimulation may influence other taste modalities beyond saltiness. For practical applications, it is important to investigate how other tastes might be affected, particularly in stroke patients, and how these changes manifest when actual food is used, which contains various forms of taste stimulation. These considerations will be addressed in future research.

## 5. Conclusions

This study highlights the effect of ETS on salt taste perception in patients with stroke. Using psychophysical experiments, it has been demonstrated that ETS significantly enhanced the perception of salt taste without changing the actual salt concentration. This suggests that ETS may help reduce daily salt intake by increasing salt taste sensitivity. However, further research and larger studies are necessary to confirm these results and to examine the wider potential of ETS in enhancing taste perception and promoting adherence to dietary interventions while monitoring blood pressure trends.

## Figures and Tables

**Figure 1 foods-13-04087-f001:**
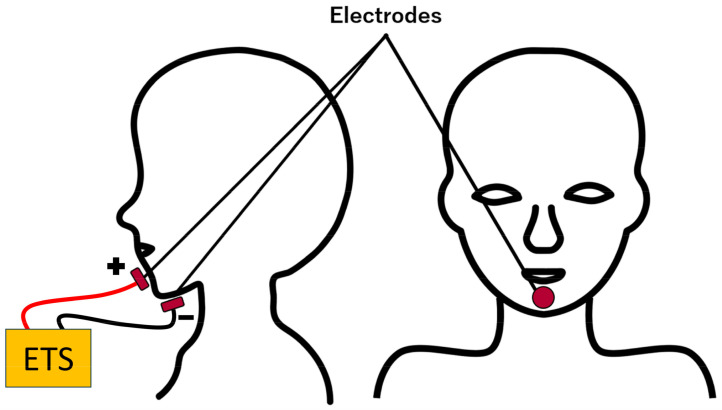
Arrangement of electrodes. Electrodes were placed on the center of the chin (anode) and the front area around the hyoid bone (cathode). During the experiment, a direct current of 1.0 mA was applied through the ETS. ETS: electrical taste stimulation.

**Figure 2 foods-13-04087-f002:**
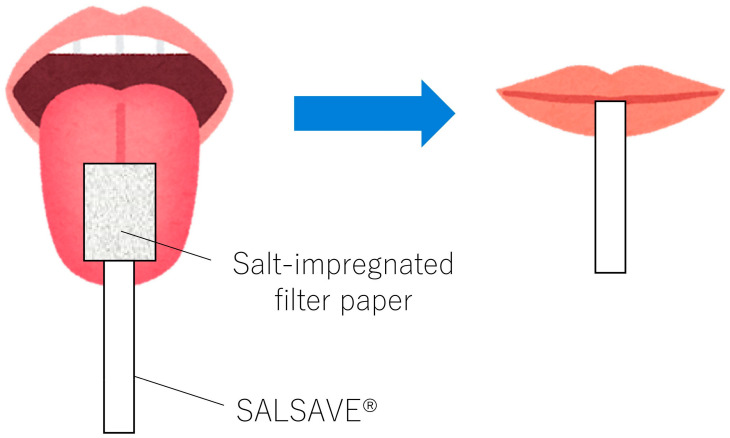
Taste examination with SALSAVE^®^. SALSAVE^®^ consists of seven filter papers, each containing different salt concentrations: 0%, 0.6%, 0.8%, 1.0%, 1.2%, 1.4%, and 1.6%. When the filter paper is placed on an individual’s tongue and their mouth is closed, the taste is typically stimulated within approximately 3 s.

**Figure 3 foods-13-04087-f003:**
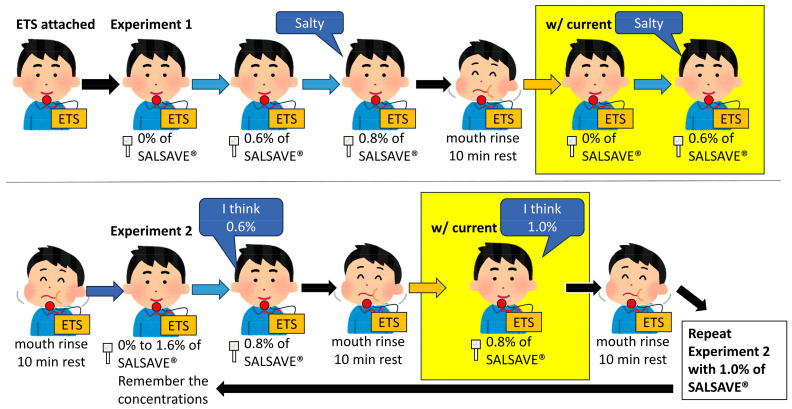
Procedure and timeline of Experiments 1 and 2. ETS: electrical taste stimulation; w/: with.

**Figure 4 foods-13-04087-f004:**
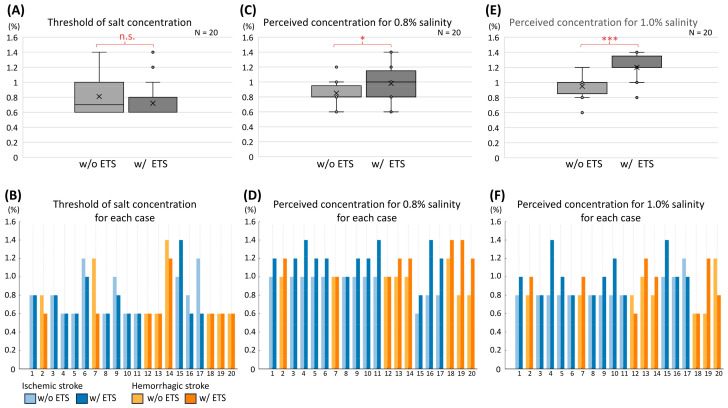
Perceived and threshold salt concentrations for Experiments 1 and 2. (**A**): Salt taste threshold without and with ETS (*p* = 0.083 by Wilcoxon signed-rank test). (**B**): Individual salt thresholds for each patient in Experiment 1. (**C**): Perceived concentration for 0.8% salt filter with ETS (*p* = 0.041 * by Wilcoxon signed-rank test). (**D**): Individual perceived concentrations for 0.8% salt filter paper in Experiment 2. (**E**): Perceived concentration for 1.0% salt filter paper with ETS (*p* < 0.001 *** by Wilcoxon signed-rank test). (**F**): Individual perceived concentrations for 1.0% salt filter paper in Experiment 2. ETS: enhanced taste sensation; n.s.: non-significant; w/: with; w/o: without.

**Table 1 foods-13-04087-t001:** Patient characteristics of the cohort.

	Median or Actual Number	IQR or %	*p* Value
Age (years old)	67.8 (mean)	13.6 (SD)	
Sex: Female	3	15%	
Presence of hypertension	14	70%	
Estimated glomerular filtration rate (mL/min/1.73 m^2^)	69.9 (mean)	18.4 (SD)	
Stroke type			
Ischemic stroke	12	60%	
Hemorrhagic stroke	8	40%	
NIHSS	3	3	
Experiment 1			
Threshold of salt concentration w/o ETS (%)	0.7%	0.4%	0.083
Threshold of salt concentration w ETS (%)	0.6%	0.2%	
The number of individuals with lowered thresholds	7	35%	
Experiment 2			
Perceived concentration for 0.8% salinity w/o ETS	0.8%	0.1%	0.041
Perceived concentration for 0.8% salinity w/ETS	1.0%	0.3%	
Number of individuals with enhanced salt perception for 0.8%	11	55%	
Perceived concentration for 1.0% salinity w/o ETS	1.0%	0.1%	<0.001
Perceived concentration for 1.0% salinity w/ETS	1.2%	0.1%	
Number of individuals with enhanced salt perception for 1.0%	17	85%	
Discomfort by ETS indicated on a numerical rating scale from 0 to 10	2	4	
No discomforts	3	15%	

Ischemic stroke included eleven cerebral infarctions and one transient ischemic attack. Hemorrhagic stroke included seven cerebral hemorrhages and one subarachnoid hemorrhage. Abbreviations. ETS: electrical taste stimulation; IQR: interquartile range; NIHSS: National Institutes of Health Stroke Scale; SD: standard deviation; w/: with, w/o: without.

## Data Availability

The original contributions presented in the study are included in the article, further inquiries can be directed to the corresponding author.

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
