# Peer review of "Enhancing Salty Taste Perception in Stroke Patients via Anodal Electrical Stimulation to the Chin"

_foods, 2024, doi:10.3390/foods13244087_

Round 1

Reviewer 1 Report

Comments and Suggestions for Authors

I have read the manuscript " Anodal Electrical Taste Stimulation to the Chin Enhances Salty Taste in Stroke Patients" with great interest. I think the subject of "improving salt perception ability through electrical stimulation" is very topical and from that perspective very interesting. However, the issues present in the manuscript cannot be ignored. The author is requested to make major revisions and address the following issues:

1.There are too many keywords, and the correlation between "hypertension", "rehabilitation", "virtual reality" and the content of the manuscript is insufficient, making them unsuitable as keywords for the manuscript. Suggest choosing more suitable keywords for replacement.

2. In section 1 Introduction, there is a lack of principles for controlling the risk of hypertension and stroke by reducing salt intake.

3. In section 1 Introduction, there is no introduction to the advantages of anodic electric taste stimulation compared to other salt reduction methods.

4. In Section 2 Materials and Methods, there is a lack of introduction to the dietary management of the subjects before the experiment. Collecting and managing the eating habits and salt intake of participants is an important prerequisite for this experiment.

5. It is recommended to merge the content of section 2.4 with section 2.1 to avoid duplication of content.

6. The innovation of this study is insufficient. And the lack of parallel experiments results in a certain degree of uncertainty in the experimental results.

7. In section 4.2, there is insufficient comparison between the two hypotheses, which makes it difficult for readers to fully understand the differences between the two hypotheses.

8. In section 4.4, the discussion on the limitations of the article is not sufficient and lacks further research directions.

9. The manuscript does not cite the latest cutting-edge research. The lack of the latest experimental results reduces the relevance and effectiveness of the paper.

10. There are a few grammar issues in the manuscript that require professional language polishing.

Comments on the Quality of English Language

The English could be improved to more clearly express the research.

Author Response

Reviewer 1

I have read the manuscript " Anodal Electrical Taste Stimulation to the Chin Enhances Salty Taste in Stroke Patients" with great interest. I think the subject of "improving salt perception ability through electrical stimulation" is very topical and from that perspective very interesting. However, the issues present in the manuscript cannot be ignored. The author is requested to make major revisions and address the following issues:

->Thank you for your kind review.

  1. There are too many keywords, and the correlation between "hypertension", "rehabilitation", "virtual reality" and the content of the manuscript is insufficient, making them unsuitable as keywords for the manuscript. Suggest choosing more suitable keywords for replacement.

->Thank you for your comments. We revised the keywords as follows;

anodal electrical taste stimulation; hypertension; rehabilitation; salt reduction; stroke; taste manipulation

  1. In section 1 Introduction, there is a lack of principles for controlling the risk of hypertension and stroke by reducing salt intake.

->Thank you for your comments. We described as follows;

High salt intake leads to hypertension by increasing blood volume through fluid retention, narrowing blood vessels, and impairing kidney function. Excess sodium activates hormonal systems like the renin-angiotensin-aldosterone system, causing further constriction of blood vessels. These combined effects result in elevated blood pressure, contributing to the development of hypertension (Page 1, Line 39-42).

  1. In section 1 Introduction, there is no introduction to the advantages of anodic electric taste stimulation compared to other salt reduction methods.

->We described as follows;

Anodal ETS has some advantages compared to other salt reduction methods: Reducing salt by replacing sodium chloride with potassium chloride cannot be used by patients with poor kidney function due to the potassium load. In addition, the cutlery-type ETS must be kept in contact with the mouth during chewing in order to be effective. (Page 2, Line 75-78)

  1. In Section 2 Materials and Methods, there is a lack of introduction to the dietary management of the subjects before the experiment. Collecting and managing the eating habits and salt intake of participants is an important prerequisite for this experiment.

->Thank you for the important suggestion. I described as follows;

Stroke patients are generally prescribed a special diet for reduced salt. The caloric intake is calculated using the Harris-Benedict equation along with an activity factor, while the salt intake is adjusted to be below 6g per day (Page 3, Line 102-104).

  1. It is recommended to merge the content of section 2.4 with section 2.1 to avoid duplication of content.

->As you suggested, we reduced the duplicated information in the Method section.

  1. The innovation of this study is insufficient. And the lack of parallel experiments results in a certain degree of uncertainty in the experimental results.

->Thank you for your important comments. As you suggested, one limitation of this study is the insufficient innovation in the research approach, which may have limited its contribution to the existing body of knowledge. Additionally, the lack of parallel experiments introduces a degree of uncertainty in the results. In future studies, we will focus on exploring more innovative methodologies and incorporate parallel experiments to enhance the reliability and depth of the findings.

  1. In section 4.2, there is insufficient comparison between the two hypotheses, which makes it difficult for readers to fully understand the differences between the two hypotheses.

->We revised section 4.2 as a whole to make them easy for readers to understand.

  1. In section 4.4, the discussion on the limitations of the article is not sufficient and lacks further research directions.

->We revised section 4.4 to discuss the lacks and further perspectives more.

  1. The manuscript does not cite the latest cutting-edge research. The lack of the latest experimental results reduces the relevance and effectiveness of the paper.

->We referred to 2 papers on ETS for stroke patients and mentioned the difference in the introduction section as follows;

Electrical taste stimulation (ETS) has been increasingly recognized for its potential in improving sensory perception, particularly in the context of reducing salt intake without compromising flavor. Previous studies have explored various forms of electrical stimulation, including electrogustometry (EGM), to assess and enhance taste perception. Electrogustometry, which involves direct stimulation of taste receptors on the tongue, has been shown to alter the perceived intensity of tastes, particularly in individuals with sensory deficits such as those following stroke. ETS, in comparison, of-ten uses anodal stimulation, which has been shown to improve sensory thresholds in various regions of the oral cavity, including the faucial pillars, which are crucial for triggering the swallowing reflex [5,6]. One of the key mechanism behind ETS is believed to involve the activation of peripheral sensory nerves, which then modulate central nervous system activity, enhancing sensory processing and potentially improving taste perception without the need for altering the chemical composition of food. This ability to modulate taste perception could be particularly beneficial for individuals seeking to reduce their salt intake, as it may allow them to enjoy food with lower sodium content while still experiencing satisfactory flavor. While the use of sour and cold sensory stimuli in stroke rehabilitation has demonstrated positive effects on swallowing funeral and pharyngeal sensory input [6], the potential of ETS in reducing salt intake without compromising flavor remains underexplored. This study aims to investigate the effects of anodal ETS on salt taste perception in stroke patients, exploring its potential as a non-invasive intervention to support salt reduction strategies in this population. (Page 2, Line 50-70)

  1. There are a few grammar issues in the manuscript that require professional language polishing.

->We had Editage.jp to revise the manuscript’s grammar. The English editing certificate was attached in non-published materials.

Reviewer 2 Report

Comments and Suggestions for Authors

The Manuscript 'Anodal Electrical Taste Stimulation to the Chin Enhances Salty Taste in Stroke Patients' investigates how anodal stimulation applied over the chin impacts the perception of salty flavour in stroke patients. The study is intriguing, and exploring alternatives to reduce salt intake in stroke patients is highly relevant. Overall, I appreciate the concept of the study and its results are quite interesting; however, I believe it is poorly presented. The introduction lacks important background information, the methods section is missing fundamental details, and the discussion feels superficial. While the study’s concept is strong, several aspects require improvement before it can be considered suitable for publication. Please see my specific comments below.

1. Firstly, the introduction would benefit from a more comprehensive background on the topic.  

1.1. It is essential to include more detailed information on prior research involving chin electrical stimulation and its effects on taste perception. A brief comparison and distinction between ETS and electrogustometry would also be beneficial to contextualise the study's methodology within established taste assessment techniques. Moreover, elaborating on the proposed mechanism of action in this context is of particular importance.  Although this is briefly mentioned in the discussion, the topic should have been properly introduced earlier to provide a foundation for the research. In the discussion, it should serve to interpret the study findings rather than to explain the technique. (Please see also my cmmt #3.2)

1.2. Clarifying the rationale behind selecting the chin as the target area may help to provide stronger reasoning for selecting this stimulation site.

1.3. I suggest incorporating these two recent and very relevant literature in the introduction: 

- Braun, T., Hamzic, S., Doerr, J.M. et al. Facilitation of oral sensitivity by electrical stimulation of the faucial pillars. Sci Rep 11, 10762 (2021). https://doi.org/10.1038/s41598-021-90262-y

- Cola, P.C., Onofri, S.M.M., Rubira, C.J. et al. Electrical, taste, and temperature stimulation in patients with chronic dysphagia after stroke: a randomized controlled pilot trial. Acta Neurol Belg 121, 1157–1164 (2021). https://doi.org/10.1007/s13760-021-01624-2

2. There is important missing methodological information that must be added to allow better understanding and replicability. 

2.1.  Concerning the electrical stimulation, details about the electrodes' size and shape should be specified. How were the electrodes attached to the chin, and what type of conductor was used to minimise skin irritation? Further, clarify how the current was administered—did it gradually ramp up and down? For instance, how long did it take to increase from zero to 1 mA? Additionally, specify the timing and duration of the stimulation. It is essential to distinguish between a constant current and a sudden 1 mA spike, as the latter could simply distract the participant from the taste stimuli.

2.2. It would help to understand the entire procedure if a timeline containing all steps and their timing was added.

2.3.  As I understand it, in both experiments 1 and 2, all participants first completed a round without stimulation, followed by evaluations with stimulation. Given the absence of a control group using sham stimulation or no stimulation at all, this presents a significant limitation for interpreting the results, as the initial evaluation round could influence responses in the subsequent round. Typically, studies randomise the order of conditions to mitigate this effect. The authors should justify why they did not randomise the order and address this as a limitation in the discussion.

2.4 The data analysis section is vague. The statement, “Fisher exact, chi-squared, Mann-Whitney U, and Wilcoxon signed-rank tests were used appropriately,” lacks specificity. Please provide detailed information regarding the type of comparisons and the statistical tests used for each data and type of comparison.

3. Results are interesting, but the current format of the discussion, overly fragmented and not in-depth, jeopardises the manuscript. Findings should be critically analysed, and their implications further explored and placed within the context of existing literature. Limitations and their significance to the research conclusions could be further explored and better articulated:

3.1. The many short subheadings (4.1, 4.2, 4.3, and 4.4) detract from the overall flow and coherence of the argument. A more integrated approach would allow for a smoother transition between ideas and better convey the significance of the findings in a holistic manner.

3.2. Some background information, which should have been included in the introduction, has been relegated to the discussion. The mechanisms behind ETS are more like an introduction to the concept rather than a discussion of the results. This misplacement hinders the discussion of the findings. The authors should add mechanisms of action in the introduction and in the discussion focus on interpreting the results in relation to existing literature, their implications, and a brief exploration of the proposed mechanisms and their relevance to the study’s findings.

3.3  While the discussion includes references to previous studies, it lacks a thorough critical analysis of the results obtained in this study. The authors should delve deeper into how their findings compare to existing literature, highlighting methodological similarities and discrepancies, e.g., population, ETS devices etc.

3.4. The limitations outlined are relevant, but they could be further elaborated. The authors should address how these limitations impact the study’s discussion and conclusions and suggest specific areas for future research. Furthermore, the point raised in cmmt #2.3 should be added.

4. A minor suggestion for making the tile more straightforward and inviting would be to rephrase it as:  "Enhancing Salty Taste Perception in Stroke Patients via Anodal Electrical Stimulation to the Chin.” - However, this is not mandatory. 

Author Response

Reviewer 2

The Manuscript 'Anodal Electrical Taste Stimulation to the Chin Enhances Salty Taste in Stroke Patients' investigates how anodal stimulation applied over the chin impacts the perception of salty flavour in stroke patients. The study is intriguing, and exploring alternatives to reduce salt intake in stroke patients is highly relevant. Overall, I appreciate the concept of the study and its results are quite interesting; however, I believe it is poorly presented. The introduction lacks important background information, the methods section is missing fundamental details, and the discussion feels superficial. While the study’s concept is strong, several aspects require improvement before it can be considered suitable for publication. Please see my specific comments below.

->Thank you for your kind review.

  1. Firstly, the introduction would benefit from a more comprehensive background on the topic.

1.1. It is essential to include more detailed information on prior research involving chin electrical stimulation and its effects on taste perception. A brief comparison and distinction between ETS and electrogustometry would also be beneficial to contextualise the study's methodology within established taste assessment techniques. Moreover, elaborating on the proposed mechanism of action in this context is of particular importance.  Although this is briefly mentioned in the discussion, the topic should have been properly introduced earlier to provide a foundation for the research. In the discussion, it should serve to interpret the study findings rather than to explain the technique. (Please see also my cmmt #3.2)

->Thank you for your important comments. We added a paragraph to describe ETS and its context.

Electrical taste stimulation (ETS) has been increasingly recognized for its potential in improving sensory perception, particularly in the context of reducing salt intake without compromising flavor. Previous studies have explored various forms of electrical stimulation, including electrogustometry (EGM), to assess and enhance taste perception. Electrogustometry, which involves direct stimulation of taste receptors on the tongue, has been shown to alter the perceived intensity of tastes, particularly in individuals with sensory deficits such as those following stroke. ETS, in comparison, of-ten uses anodal stimulation, which has been shown to improve sensory thresholds in various regions of the oral cavity, including the faucial pillars, which are crucial for triggering the swallowing reflex [5,6]. One of the key mechanism behind ETS is believed to involve the activation of peripheral sensory nerves, which then modulate central nervous system activity, enhancing sensory processing and potentially improving taste perception without the need for altering the chemical composition of food. This ability to modulate taste perception could be particularly beneficial for individuals seeking to reduce their salt intake, as it may allow them to enjoy food with lower sodium content while still experiencing satisfactory flavor. While the use of sour and cold sensory stimuli in stroke rehabilitation has demonstrated positive effects on swallowing funeral and pharyngeal sensory input [6], the potential of ETS in reducing salt intake without compromising flavor remains underexplored. This study aims to investigate the effects of anodal ETS on salt taste perception in stroke patients, exploring its potential as a non-invasive intervention to support salt reduction strategies in this population. (Page 2, Line 50-70)

1.2. Clarifying the rationale behind selecting the chin as the target area may help to provide stronger reasoning for selecting this stimulation site.

->Thank you for your important comments. We added a paragraph to describe ETS and its context in method sections as follows.

Methods

Previous studies have demonstrated that electrical taste stimulation with the jaw as the anode and the neck as the cathode enhances saltiness in healthy individuals. This method, which does not require placing electrodes or cables inside the mouth, is expected to enable virtual taste enhancement without interfering with eating. Additionally, stimulation with the region around the chin as the anode and the region around the jaw as the cathode is presumed to activate taste-related nerves by directing current into the oral cavity. Using this approach, we have previously reported enhanced saltiness in stroke patients. Based on these findings, this study employed electrical taste stimulation with the region around the chin as the anode and the region around the jaw as the cathode. (Page 3, Line 133-141)

1.3. I suggest incorporating these two recent and very relevant literature in the introduction:

- Braun, T., Hamzic, S., Doerr, J.M. et al. Facilitation of oral sensitivity by electrical stimulation of the faucial pillars. Sci Rep 11, 10762 (2021). https://doi.org/10.1038/s41598-021-90262-y

- Cola, P.C., Onofri, S.M.M., Rubira, C.J. et al. Electrical, taste, and temperature stimulation in patients with chronic dysphagia after stroke: a randomized controlled pilot trial. Acta Neurol Belg 121, 1157–1164 (2021). https://doi.org/10.1007/s13760-021-01624-2

->Thank you for your constructive suggestion. We referred to these articles in the introduction as described as the response of 1.1.

  1. There is important missing methodological information that must be added to allow better understanding and replicability.

2.1.  Concerning the electrical stimulation, details about the electrodes' size and shape should be specified. How were the electrodes attached to the chin, and what type of conductor was used to minimise skin irritation? Further, clarify how the current was administered—did it gradually ramp up and down? For instance, how long did it take to increase from zero to 1 mA? Additionally, specify the timing and duration of the stimulation. It is essential to distinguish between a constant current and a sudden 1 mA spike, as the latter could simply distract the participant from the taste stimuli.

->We described as follows;

After the third week of admission, the patients were transferred to the general ward. Similar to a previous study, two psychophysical experiments as described later were conducted to demonstrate the effect of ETS on taste perception [6]. As shown in Figure 1, the disposable electrodes, 36 mm in diameter, equipped with conductive gel and metal pins for optimal contact (C915F30, Saitama, Chibara, Japan), were placed at the center of the chin (anode) and in the area around the hyoid bone (cathode). These electrodes were directly adhered to the chin using adhesive to minimize skin irritation. The stimulation wires were standard copper wires (0.2 Sq) connected to the electrodes with crocodile clips. The ETS device is equipped with an ampere meter, and the circuit is designed to allow a constant current to flow using a direct current power supply. The current was gradually increased from 0 mA to 1 mA over a period of about 2 seconds manually and allowed to stabilize before the experiment began. The current was continuously applied throughout the experiment. A constant current was used, rather than a sudden 1 mA spike, to avoid distracting the participants from the taste stimuli. (Page 3, Line 118-131)

2.2. It would help to understand the entire procedure if a timeline containing all steps and their timing was added.

->Thank you for your important suggestion. We added Figure 3 to describe the timeline.

2.3.  As I understand it, in both experiments 1 and 2, all participants first completed a round without stimulation, followed by evaluations with stimulation. Given the absence of a control group using sham stimulation or no stimulation at all, this presents a significant limitation for interpreting the results, as the initial evaluation round could influence responses in the subsequent round. Typically, studies randomise the order of conditions to mitigate this effect. The authors should justify why they did not randomise the order and address this as a limitation in the discussion.

->We described as follows;

A potential limitation of this study is the lack of randomization in the order of conditions and the absence of a sham stimulation control group. This could introduce a carryover effect from the initial evaluation to the subsequent round with stimulation. Randomizing the order and including a sham group would have strengthened the design and minimized bias. (Page 9, Line 327-333).

2.4 The data analysis section is vague. The statement, “Fisher exact, chi-squared, Mann-Whitney U, and Wilcoxon signed-rank tests were used appropriately,” lacks specificity. Please provide detailed information regarding the type of comparisons and the statistical tests used for each data and type of comparison.

->We described as follows;

Fisher’s exact test and chi-squared test were used to compare the proportion of the categorical variables. Mann–Whitney U test was used to compare continuous and ordinal variables in independent groups. Wilcoxon signed-rank test was used as a paired test to compare continuous and ordinal variables. These analyses were conducted using SPSS v28.0.0 (IBM, NY, USA), Python v3.9, and matplotlib v3.9.2. Statistical significance was set at p < 0.05. (Page 5, Line 183-190).

  1. Results are interesting, but the current format of the discussion, overly fragmented and not in-depth, jeopardises the manuscript. Findings should be critically analysed, and their implications further explored and placed within the context of existing literature. Limitations and their significance to the research conclusions could be further explored and better articulated:

3.1. The many short subheadings (4.1, 4.2, 4.3, and 4.4) detract from the overall flow and coherence of the argument. A more integrated approach would allow for a smoother transition between ideas and better convey the significance of the findings in a holistic manner.

->Thank you for your comments. We revised the whole manuscript and integrated each paragraph.

3.2. Some background information, which should have been included in the introduction, has been relegated to the discussion. The mechanisms behind ETS are more like an introduction to the concept rather than a discussion of the results. This misplacement hinders the discussion of the findings. The authors should add mechanisms of action in the introduction and in the discussion focus on interpreting the results in relation to existing literature, their implications, and a brief exploration of the proposed mechanisms and their relevance to the study’s findings.

->Thank you for your constructive suggestion. We referred to the papers you suggested in the 1.3. The introduction and discussion each mentioned the mechanism of ETS. This is because the ETS device in the introduction and our device are different, and we thought it necessary to discuss them separately.

3.3  While the discussion includes references to previous studies, it lacks a thorough critical analysis of the results obtained in this study. The authors should delve deeper into how their findings compare to existing literature, highlighting methodological similarities and discrepancies, e.g., population, ETS devices etc.

->Thank you for your valuable comment. We agree that a more thorough critical analysis of our findings in relation to existing literature would be beneficial. However, given the novelty of using transcutaneous electrical taste stimulation (ETS) in stroke patients, there are no prior studies that specifically investigate this approach in this population. As such, a detailed comparison with previous research regarding methodological aspects such as population and ETS devices is not feasible at this stage. We plan to address these comparisons and provide a more in-depth analysis in future studies as we expand the understanding of ETS in stroke rehabilitation.

3.4. The limitations outlined are relevant, but they could be further elaborated. The authors should address how these limitations impact the study’s discussion and conclusions and suggest specific areas for future research. Furthermore, the point raised in cmmt #2.3 should be added.

->We described as follows;

A potential limitation of this study is the lack of randomization in the order of conditions and the absence of a sham stimulation control group. This could introduce a carryover effect from the initial evaluation to the subsequent round with stimulation. Randomizing the order and including a sham group would have strengthened the design and minimized bias. (Page 9, Line 327-331).

  1. A minor suggestion for making the tile more straightforward and inviting would be to rephrase it as: "Enhancing Salty Taste Perception in Stroke Patients via Anodal Electrical Stimulation to the Chin.” - However, this is not mandatory.

->Thank you for your suggestion. We changed as you suggested.

Reviewer 3 Report

Comments and Suggestions for Authors

This manuscript presents a prospective study on the effect of anodal electrical taste stimulation (ETS) on salt taste perception in stroke patients. Reducing salt intake is crucial for controlling hypertension and preventing stroke, but low salt foods often lack flavor. The ETS technology discussed in the article provides a possible way to solve this problem and has potential application value in improving the dietary management and blood pressure control of stroke patients However, there are several areas that need improvement.

Major point:

1. The sample size of 20 patients is relatively small, which may limit the generalizability of the results. The authors should discuss the potential impact of this small sample size on the validity of their findings and consider whether a larger sample size would be beneficial in future studies.

2. The two experiments used to assess salt taste perception seem appropriate, but the methods could be described in more detail. For example, it would be helpful to have more information about how the salt-impregnated filter papers were prepared and standardized,and how to set ETS device parameters .

3. page3,line103--The use of a single current intensity (1.0 mA) for ETS may not be sufficient to fully explore the dose-response relationship. Future studies could consider testing different current intensities to determine the optimal stimulation parameters.

4. page6,line185--age, sex, presence of hypertension, eGFR, NIHSS, and discomfort between patients,specific statistical data should be provided to demonstrate that these influencing factors are not statistically significant.

5. While the results suggest that ETS may enhance salt taste perception in stroke patients, the statistical significance of some of the findings is marginal(p = 0.041).(line165) The authors should be cautious in interpreting these results and consider conducting additional analyses or replicating the study with a larger sample size to confirm their findings.

6. The impact of this study on clinical practice and future research can be further elaborated. For example, the author can discuss how ETS can be integrated into the dietary management plan of stroke patients, whether the device is convenient for practical use without increasing the cost of use for patients, and what further research is needed to optimize its use.

Miner point

1. The photos in Fig. 1and Fig.2 has been used in your previous study (Journal of Robotics and Mechatronics 2021, 33, 11281134, doi:10.20965), you should retake photos.

Author Response

Reviewer 3

This manuscript presents a prospective study on the effect of anodal electrical taste stimulation (ETS) on salt taste perception in stroke patients. Reducing salt intake is crucial for controlling hypertension and preventing stroke, but low salt foods often lack flavor. The ETS technology discussed in the article provides a possible way to solve this problem and has potential application value in improving the dietary management and blood pressure control of stroke patients However, there are several areas that need improvement.

->Thank you for your kind review.

Major point:

  1. The sample size of 20 patients is relatively small, which may limit the generalizability of the results. The authors should discuss the potential impact of this small sample size on the validity of their findings and consider whether a larger sample size would be beneficial in future studies.

->Thank you for your important comment. We described this in the limitation section as follows;

The sample size was as small as 20 patients, and the statistical siginificance of some of the findings is marginal (p=0.041). Therefore, we should confirm ETS effects in a larger sample size. (Page 9, Line 318-319)

  1. The two experiments used to assess salt taste perception seem appropriate, but the methods could be described in more detail. For example, it would be helpful to have more information about how the salt-impregnated filter papers were prepared and standardized,and how to set ETS device parameters .

->Thank you for your comments. We described the methods section about the use of salt-impregnated filter papers called SALSAVE. This is a product for taste perceive test for human and widely used, leading to the standardized procedure as follows;

First, starting with the paper from the lowest concentration to higher concentration in order, the paper that elicited a salty taste was recorded as the taste threshold. Followed by a thorough mouth rinse and a rest period of at least 10 min, the same procedure was repeated using a direct current of 1.0 mA via the ETS and the salt taste threshold was re-evaluated. Experiment 1 was conducted without disclosing salt concentrations to the patients.

Experiment 2 was conducted after the patients thoroughly rinsed their mouth and rested for at least 10 min. The seven concentration levels of SALSAVE® were then dis-closed to the patients and they were required to remember them. The patients were then given 0.8% filter paper without revealing the concentration. They were asked to determine the concentration of the filter papers ranging from 0.6% to 1.6% without ETS stimulation. After rinsing their mouth and resting for at least 10 min, the same filter papers were given to the participants with a 1.0 mA direct current applied by the ETS. The patients were then asked to indicate the concentration they assumed it may be. The patients were then provided with 1.0% filter paper and the same procedure was re-peated. This experiment was conducted with and without ETS at two different concentrations: 0.8% and 1.0%.

The primary evaluation items were the proportions of whether the ETS device decreased the threshold concentration in Experiment 1 and whether the perceived concentration was increased by the ETS in Experiment 2. (Page 4, Line 156- Page 5, 176)

The ETS device parameters are described as follows;

After the third week of admission, the patients were transferred to the general ward. Similar to a previous study, two psychophysical experiments were conducted to demonstrate the effect of ETS on taste perception [6]. As shown in Figure 1, the elec-trodes (11B3X10035000001, Saitama, Chibara, Japan) were placed at the center of the chin (anode) and in the area around the hyoid bone (cathode). The ETS device is equipped with an ampere meter, and the circuit is designed to allow a constant current to flow using a direct current power supply. (Page 3, Line 118-131)

  1. page3,line103--The use of a single current intensity (1.0 mA) for ETS may not be sufficient to fully explore the dose-response relationship. Future studies could consider testing different current intensities to determine the optimal stimulation parameters.

->We described as follows in the limitation section;

The use of a single current intensity (1.0 mA) in this study may not be sufficient to fully explore the dose-response relationship. Future studies should consider testing a range of current intensities to determine the optimal stimulation parameters. (Page 9, Line 322-325)

  1. page6,line185--age, sex, presence of hypertension, eGFR, NIHSS, and discomfort between patients,specific statistical data should be provided to demonstrate that these influencing factors are not statistically significant.

->We described as follows;

In Experiments 1 and 2, age, sex, presence of hypertension, eGFR, NIHSS score, and discomfort were compared between patients with and without ETS effects. There were not statistically significant differences between the two groups (p = 0.341, 0.207, 0.999, 0.843, 0.547, 0.678, respectively in Experiment 1. p = 0.552, 0.465, 0.785, 0.764, 0.999, 0.557, respectively in Experiment 2 with 0.8% salt filter paper. p = 0.842, 0.779, 0.765, 0.473, 0.237, 0.337, respectively in Experiment 2 with 1.0% salt filter paper). (Page 7, Line 242-247)

  1. While the results suggest that ETS may enhance salt taste perception in stroke patients, the statistical significance of some of the findings is marginal(p = 0.041).(line165) The authors should be cautious in interpreting these results and consider conducting additional analyses or replicating the study with a larger sample size to confirm their findings.

->We described as follows;

The sample size was as small as 20 patients, and the statistical siginificance of some of the findings is marginal (p=0.041). Therefore, we should confirm ETS effects in a larger sample size. (Page 9, Line 318-320)

  1. The impact of this study on clinical practice and future research can be further elaborated. For example, the author can discuss how ETS can be integrated into the dietary management plan of stroke patients, whether the device is convenient for practical use without increasing the cost of use for patients, and what further research is needed to optimize its use.

->We described as follows;

Further research is needed to integrate ETS into the dietary management plans of stroke patients and to evaluate ETS's long-term practicality and cost-effectiveness to ensure it is both beneficial and accessible for stroke patients. (Page 9, Line 333-336)

Miner point

  1. The photos in Fig. 1and Fig.2 has been used in your previous study (Journal of Robotics and Mechatronics 2021, 33, 1128–1134, doi:10.20965), you should retake photos.

->Thank you for your pointing out. The figures seem similar, but our figures were truly made by ourselves and were not used in the previous article. They are original.

Round 2

Reviewer 2 Report

Comments and Suggestions for Authors

I thank the authors for their review. Most of my comments were addressed; however, I have some follow-up comments to clarify the points that still need improvement. I have kept the comment numbering consistent with the original review to make it easier to follow up.

1.1 Regarding the added paragraph on ETS, some parts of it do not contain reference citations, specifically:

 "Electrical taste stimulation (ETS) has been increasingly recognized for its potential in improving sensory perception, particularly in the context of reducing salt intake without compromising flavor.” 

 "Previous studies have explored various forms of electrical stimulation, including electrogustometry (EGM), to assess and enhance taste perception“ 

Please add citations to those statements

1.2  Similarly, clarification on the positioning of electrodes needs attention. The authors added the following in the methods:

 "Previous studies have demonstrated that electrical taste stimulation with the jaw as the anode and the neck as the cathode enhances saltiness in healthy individuals.”  

However, there is again no citation to this claim; please do so. 

Furthermore, the phrase 

"Additionally, stimulation with the region around the chin as the anode and the region around the jaw as the cathode is presumed to (…)” 

is a bit unconventional. It would be clearer to adjust this to: 

"Additionally, placing the anode over the chin and the cathode over the jaw is presumed to (…)”

2.1 I asked about the conductor used to minimize skin irritation due to the electric current. After the authors added further details, I believe the "conductor to minimize skin irritation” is the conductive gel… Therefore, the phrase:

"These electrodes were directly adhered to the chin using adhesive to minimize skin irritation.”

Seems incorrect. I suggest keeping how the electrodes were adhered to the skin but removing the part of skin irritation, which is misplaced here.

2.4 Perhaps I wasn’t clear in my previous comment. When I asked the authors to specify which test was used for each data analysis, I meant, for example, in Experiment 2, where the perceived concentrations are compared (e.g., for 0.8% and 1.0% salt filter papers with and without ETS). What was the specific test used for this analysis?

Please explicitly state in the statistical analysis section which test was used for each dependent measure. This will help clearly link each test to the measures and comparisons analyzed.

3.1 I haven’t seen any changes that make integration and flow between the end of a subheading and the following one. A more integrated approach would allow for a smoother transition between ideas and better convey the significance of the findings holistically.

3.3. It might be relevant to integrate the author's reply to my comment in the discussion section. i.e., "there are no prior studies that specifically investigate this approach in this population. As such, a detailed comparison with previous research regarding methodological aspects such as population and ETS devices is not feasible at this stage. We plan to address these comparisons and provide a more in-depth analysis in future studies as we expand the understanding of ETS in stroke rehabilitation."

Author Response

Reviewer 2

I thank the authors for their review. Most of my comments were addressed; however, I have some follow-up comments to clarify the points that still need improvement. I have kept the comment numbering consistent with the original review to make it easier to follow up.

->Thank you for the kind review. We have revised our manuscript according to the reviewer’s suggestions.

1.1 Regarding the added paragraph on ETS, some parts of it do not contain reference citations, specifically:

 "Electrical taste stimulation (ETS) has been increasingly recognized for its potential in improving sensory perception, particularly in the context of reducing salt intake without compromising flavor.”

->We added the references as follows;

  1. Katsuki, M.; Fukushima, T.; Goto, T.; Hanaoka, Y.; Wada, N.; Nakamura, T.; Sasaki, S.; Horiuchi, T. Anodal Electrical Taste Stimulation to the Chin Enhances the Salt Taste Perception in Subarachnoid Hemorrhage Patients. Cureus 2024, doi:10.7759/cureus.56630.
  2. Funamizu, T.; Matsumoto, R.; Suzuki, A.; Watabe, K.; Nakamura, H.; Kasamatsu, C. Sensory Studies on the Taste and Flavor Perception of Food Products by Anodal Transcutaneous Electrical Stimulation. Hypertension Research 2024, doi:10.1038/s41440-024-01867-5.
  3. Ullah, A.; Liu, Y.; Wang, Y.; Gao, H.; Wang, H.; Zhang, J.; Li, G. E-Taste: Taste Sensations and Flavors Based on Tongue’s Electrical and Thermal Stimulation. Sensors (Basel) 2022, 22, doi:10.3390/s22134976.

 "Previous studies have explored various forms of electrical stimulation, including electrogustometry (EGM), to assess and enhance taste perception“

->We added the references as follows;

  1. Dutta, T.M.; Josiah, A.F.; Cronin, C.A.; Wittenberg, G.F.; Cole, J.W. Altered Taste and Stroke: A Case Report and Literature Review. Top Stroke Rehabil 2013, 20, 78–86, doi:10.1310/tsr2001-78.
  2. Ohla, K.; Hudry, J.; le Coutre, J. The Cortical Chronometry of Electrogustatory Event-Related Potentials. Brain Topogr 2009, 22, 73–82, doi:10.1007/s10548-009-0076-7.
  3. McClure, S.T.; Lawless, H.T. A Comparison of Two Electric Taste Stimulation Devices. Physiol Behav 2007, 92, 658–664, doi:10.1016/j.physbeh.2007.05.010.

1.2  Similarly, clarification on the positioning of electrodes needs attention. The authors added the following in the methods:

 "Previous studies have demonstrated that electrical taste stimulation with the jaw as the anode and the neck as the cathode enhances saltiness in healthy individuals.” 

However, there is again no citation to this claim; please do so.

->We added the references as follows;

  1. Nakamura, H.; Mizukami, M.; Aoyama, K. Method of Modifying Spatial Taste Location through Multielectrode Galvanic Taste Stimulation. IEEE Access 2021, 9, 47603–47614, doi:10.1109/ACCESS.2021.3068263.

Furthermore, the phrase

"Additionally, stimulation with the region around the chin as the anode and the region around the jaw as the cathode is presumed to (…)”

is a bit unconventional. It would be clearer to adjust this to:

"Additionally, placing the anode over the chin and the cathode over the jaw is presumed to (…)”

->We changed as you suggested.

Additionally, placing the anode over the chin and the cathode over the jaw is presumed to activate taste-related nerves by directing current into the oral cavity. (Page 3, Line 136-138).

2.1 I asked about the conductor used to minimize skin irritation due to the electric current. After the authors added further details, I believe the "conductor to minimize skin irritation” is the conductive gel… Therefore, the phrase:

"These electrodes were directly adhered to the chin using adhesive to minimize skin irritation.”

Seems incorrect. I suggest keeping how the electrodes were adhered to the skin but removing the part of skin irritation, which is misplaced here.

->We rephrased as follows;

These electrodes were directly adhered to the chin using the adhesive (gel). (Page 3, Line 123-124 )

2.4 Perhaps I wasn’t clear in my previous comment. When I asked the authors to specify which test was used for each data analysis, I meant, for example, in Experiment 2, where the perceived concentrations are compared (e.g., for 0.8% and 1.0% salt filter papers with and without ETS). What was the specific test used for this analysis?

Please explicitly state in the statistical analysis section which test was used for each dependent measure. This will help clearly link each test to the measures and comparisons analyzed.

->We described in the method section as follows;

The Shapiro–Wilk test confirmed a normal distribution. Fisher’s exact test and chi-squared test were used to compare the proportion of the categorical variables. Mann–Whitney U test was used to compare continuous and ordinal variables in independent groups. Wilcoxon signed-rank test was used as a paired test to compare continuous and ordinal variables. The outcomes in experiments 1 and 2 were evaluated using the Wilcoxon signed-rank test (Page 5, Line 185-191).

We also described the test names next to the p values in the result sections.

3.1 I haven’t seen any changes that make integration and flow between the end of a subheading and the following one. A more integrated approach would allow for a smoother transition between ideas and better convey the significance of the findings holistically.

->Thank you very much for your very constructive comments. We did our best, but this time, the discussion paragraphs ended up just being evaluated from various perspectives, and the flow was not emphasized. As you say, it is essential to emphasize the flow when constructing it, and we will do so in the next large-scale sample test and its report. We apologize for not meeting your expectations.

 3.3. It might be relevant to integrate the author's reply to my comment in the discussion section. i.e., "there are no prior studies that specifically investigate this approach in this population. As such, a detailed comparison with previous research regarding methodological aspects such as population and ETS devices is not feasible at this stage. We plan to address these comparisons and provide a more in-depth analysis in future studies as we expand the understanding of ETS in stroke rehabilitation."

->Thank you for your constructive suggestions. We added the sentences as follows;

There are no prior studies that specifically investigate this approach in this population. As such, a detailed comparison with previous research regarding methodological aspects such as population and ETS devices is not feasible at this stage. We plan to address these comparisons and provide a more in-depth analysis in future studies as we expand the understanding of ETS in stroke rehabilitation. (Page 9, Line 345- Page 10, Line 349 )

Reviewer 3 Report

Comments and Suggestions for Authors

The authors have addressed most of my questions.

Author Response

->Thank you for the kind review.